# Myostatin Inhibition-Induced Increase in Muscle Mass and Strength Was Amplified by Resistance Exercise Training, and Dietary Essential Amino Acids Improved Muscle Quality in Mice

**DOI:** 10.3390/nu13051508

**Published:** 2021-04-29

**Authors:** Jiwoong Jang, Sanghee Park, Yeongmin Kim, Jiyeon Jung, Jinseok Lee, Yewon Chang, Sang Pil Lee, Bum-Chan Park, Robert R. Wolfe, Cheol Soo Choi, Il-Young Kim

**Affiliations:** 1Korea Mouse Metabolic Phenotyping Center, Lee Gil Ya Cancer and Diabetes Institute, Gachon University, Incheon 21999, Korea; korea81@hanmail.net (J.J.); sangheepark1@gachon.ar.kr (S.P.); 2Gil Medical Center, Department of Internal Medicine, Gachon University, Incheon 21565, Korea; 3Department of Molecular Medicine, College of Medicine, Gachon University, Incheon 21565, Korea; 4Department of Health Sciences and Technology, GAIHST, Gachon University, Incheon 21999, Korea; 1893kin@naver.com (Y.K.); jyjung031@gmail.com (J.J.); jinseok0515@gmail.com (J.L.); jjangye1@gachon.ac.kr (Y.C.); 5Y-Biologics, Inc. 29 Techno 4-ro, 3rd FL., Yuseong-gu, Daejeon 34014, Korea; info@ybiologics.com (S.P.L.); parkb2@ybiologics.com (B.-C.P.); 6The Center for Translational Research in Aging & Longevity, Department of Geriatrics, Donald W. Reynolds Institute on Aging, University of Arkansas for Medical Sciences, Little Rock, AR 72205, USA; RWolfe2@uams.edu

**Keywords:** soluble activin receptor type IIB, protein turnover, mass spectrometry, deuterium oxide, essential amino acids, resistance exercise training

## Abstract

It has been frequently reported that myostatin inhibition increases muscle mass, but decreases muscle quality (i.e., strength/muscle mass). Resistance exercise training (RT) and essential amino acids (EAAs) are potent anabolic stimuli that synergistically increase muscle mass through changes in muscle protein turnover. In addition, EAAs are known to stimulate mitochondrial biogenesis. We have investigated if RT amplifies the anabolic potential of myostatin inhibition while EAAs enhance muscle quality through stimulations of mitochondrial biogenesis and/or muscle protein turnover. Mice were assigned into ACV (myostatin inhibitor), ACV+EAA, ACV+RT, ACV+EAA +RT, or control (CON) over 4 weeks. RT, but not EAA, increased muscle mass above ACV. Despite differences in muscle mass gain, myofibrillar protein synthesis was stimulated similarly in all vs. CON, suggesting a role for changes in protein breakdown in muscle mass gains. There were increases in MyoD expression but decreases in Atrogin-1/MAFbx expression in ACV+EAA, ACV+RT, and ACV+EAA+RT vs. CON. EAA increased muscle quality (e.g., grip strength and maximal carrying load) without corresponding changes in markers of mitochondrial biogenesis and neuromuscular junction stability. In conclusion, RT amplifies muscle mass and strength through changes in muscle protein turnover in conjunction with changes in implicated signaling, while EAAs enhance muscle quality through unknown mechanisms.

## 1. Introduction

Skeletal muscle is not only the largest organ in the body that directly affects one’s physical function but the largest amino acid reservoir that provides essential amino acids (EAA) to other critical tissues/organs for synthesis of new proteins for various purposes, thereby affecting one’s health and quality of life. Thus, a progressive loss of muscle mass over time that leads to muscle wasting often seen in clinical conditions such as cancer cachexia and sarcopenia can increase morbidity, including contributing to insulin resistance, diabetes, and obesity [1,2]. Consequently, research efforts have been dedicated to developing effective therapeutics against muscle wasting. Unfortunately, efficacious and safe drugs have not yet been developed [3,4].

Myostatin is a growth and differentiation factor-8 (GDF-8) cytokine primarily expressed and secreted in skeletal muscles [5,6]. Myostatin acts as a negative regulator of muscle mass. Briefly, myostatin binds to the activin receptor type IIB (ActRIIB) to activate SMAD2/3 and inhibit the expression of MyoD, which plays an important role in differentiation and repair of skeletal muscle [7,8]. Myostatin also inhibits the Akt1/mTORC1 axis, which leads to the inhibition of protein synthesis. Translocation of FoxOs to the nucleus is also induced by myostatin via inhibition of Akt1, resulting in the activation of the transcription of muscle specific ubiquitin E3 ligases [9], thereby resulting in the acceleration of protein breakdown. Over-expression of myostatin induces muscle wasting by inhibiting Akt1/mTORC1 signaling in vivo [10], and decreases myotube diameter through increased atrogin-1 and decreased MyoD expression in vitro [11]. Conversely, it was shown that inhibition of myostatin leads to muscle mass gain. For example, myostatin antisense RNA reduced muscle wasting through decreased atrogin-1 and increased MyoD mRNA expression in tumor-bearing mice [12]. In addition, it was shown that a 4-week treatment of an anti-myostatin antibody induced gains in muscle mass and strength in young and old mice [13]. Further, it appears that muscle growth induced by myostatin inhibition is attributed in part through increased activation and proliferation of satellite cells [14]. Taken together, these results indicate myostatin is a critical regulator that limits gains in muscle mass through changes in both rates of protein synthesis and breakdown. Inhibition of myostatin secretion can thus increase skeletal muscle mass through various mechanisms.

However, it was frequently reported that inhibition of myostatin decreases muscle quality [15,16,17], defined here as the capacity to generate force at a given muscle mass or volume (strength/muscle mass). The underlying mechanisms responsible for decreased muscle quality in response to myostatin inhibition are unknown. Potential mechanisms may include suppression of mitochondrial biogenesis and/or protein turnover as a (in)direct result of myostatin inhibition. Consistent with this notion, it was demonstrated that deficiency of myostatin decreases the protein expression of PGC1α, a central regulator in mitochondrial biogenesis [18]. The resulting depletion of mitochondria was accompanied by reductions in specific force generation (muscle quality) in mice [19]. Furthermore, inhibition of myostatin in C2C12 cells treated with dexamethasone appears to suppress protein breakdown through the inhibition of ubiquitin proteasome system (UPS) without affecting the rate of protein synthesis, as determined by the surfacing sensing of translation (SUnSET) method [20]. These results suggest that impaired muscle quality resulting from myostatin inhibition may be due at least in part to decreases in mitochondrial biogenesis and/or muscle protein turnover.

Resistance exercise training (RT) and essential amino acids (EAAs) are two physiological stimuli that lead to stimulation of muscle protein turnover (i.e., replacement of old, nonfunctional with new, functional proteins). Both stimuli increase muscle protein synthesis (MPS) to a rate greater than the rate of muscle protein breakdown (MPB) [21,22], and/or activate mitochondrial biogenesis (mainly through EAAs). In addition to an independent effect of each treatment [23,24,25], combination of RT and EAAs synergistically increases net muscle protein synthesis [26]. In a long-term study, combined treatment of RT and EAAs improved muscle mass and muscle quality (strength/mass) in humans [27]. Furthermore, it was shown that EAAs, mainly through its components branched chain amino acids (BCAAs), increases mitochondrial biogenesis and muscle function in middle-aged mice [28] and humans [29]. In addition, it seems that the stability of neuromuscular junction (NMJ) affects muscle strength [30] likely though altering neural transmission, implying its potential role for muscle quality. 

Taken together, these previous results have led us to hypothesize that 1) RT would increase muscle mass and strength above that induced by ACV, which will be further amplified by EAAs, and 2) EAAs alone would increase muscle quality in muscle hypertrophied by ACV. To test the hypotheses, we employed both molecular biology and stable-isotope tracer-based flux approaches.

## 2. Materials and Methods

### 2.1. Animal Care

Seven-weeks old C57BL/6J male mice were purchased from Daehan Bio Link (DBL, Korea). Mice per cage were housed in a climate-controlled room at 23 °C with a 12-h light/dark cycle with ad libitum access to food and water during the entire experimental period. Animal experimental procedures were carried out with the approval of the Center of Animal Care and Use Facility of the Gachon University Lee Gil Ya Cancer and Diabetes Institute. The protocol was approved by the Committee on the Ethics of Animal Experiments of the Gachon University Lee Gil Ya Cancer and Diabetes Institute (permit number: LCDI-2019-0010).

### 2.2. Experimental Design

The mice were assigned to sedentary control group (CON, *n* = 5), myostatin inhibition (sActRIIB-Fc) only (ACV, *n* = 5), sActRIIB-Fc with RT (ACV+RT, *n* = 5), sActRIIB-Fc with EAAs (ACV+EAA, *n* = 5), or sActRIIB-Fc with RT and EAAs (ACV+EAA+RT, *n*= 4) (one died in the ACV+EAA+RT group during the experiment). The sample size was calculated using G*power software version 3.1.9.7 (Heinrich Heine University, Düsseldorf, Germany). The sample size was chosen as 25 (5 mice/group) in order to detect a minimum difference between five groups (α = 0.05, β = 0.80). Mice were housed individually. The sActRIIB-Fc (Y-Biologics, Korea) was given as intraperitoneal (i.p.) injections once per week for four weeks at a dose of 20 mg/kg, whereas control mice were injected with an equivalent volume of saline for 4 weeks. EAAs were orally administered twice a day (1.5 g/kg) at 9:00 A.M. and 5:00 P.M. and the detailed composition is presented in Table 1. On RT days, supplement was taken immediately after each session of RT.

### 2.3. Ladder Climbing Exercise Protocol

RT involved mice performing ladder climbing three times per week for 4 weeks utilizing a 1-m ladder with rungs that were 15 mm apart and inclined at 85°. Prior to the RT, familiarization exercise sessions without and with load (second and third day) for 3 days were conducted. During each RT session, mice climbed the ladder with 50%, 70%, 90%, and 100% of the previous maximal carrying load (or their body weight during the first session of the RT), and the load was increased by 3 g for each subsequent repetition up to ten repetitions. Once the mice reached to the top of the ladder, they rested for 2 min in the house space on the top of the ladder apparatus (modified from Hornberger and Farrar) [31].

### 2.4. Four Limb Grip Strength Test

Four-limb grip strength was measured using a grip strength meter (DEFII-002, Chatillon, Largo, FL, USA) after 4 weeks of intervention. Briefly, after confirming that the mice held the wire grid with their four limbs, they were gently pulled back by their tail until they were completely separated from the grid. Measured force was recorded for five times. The maximal force was used for analysis [32].

### 2.5. 2H_2_O Labeling Protocol

Mice were administered with an i.p. bolus of 35 mL/kg of 99% ^2^H_2_O as previously described [33], followed by ad libitum access to drinking water enriched to 8% ^2^H_2_O for the duration of the study.

### 2.6. Sample Collection

After 4 weeks of experiment, all animals were anesthetized with avertin and sacrificed by cervical dislocation. Then, hindlimb muscle (soleus, plantaris, gastrocnemius, tibialis anterior, extensor didigitorum longus, and quadriceps femoris) samples were excised and immediately frozen in liquid nitrogen and stored at −80 °C for further analysis.

### 2.7. Isolation of Myofibrilar and Mitochondrial Subfractions

Myofibrillar and mitochondrial subfractions were extracted by differential centrifugation method following glass homogenization as previously described [34]. Briefly, the muscles of interest were homogenized in a buffer consisting of 550 mM KCl, 5 mM EGTA and 100 mM MOPS and centrifuged at 800 G for 5 min at 4 °C for extracting myofibrillar fraction. The remaining supernatant was centrifuged further at 7000 G for 10 min at 4 °C for isolating mitochondrial fraction.

### 2.8. Body Water and Muscle Protein Enrichment Analysis

Water and alanine enrichment were analyzed using GC-MS (Models 7890A/5975; Agilent Technologies, Santa Clara, CA, USA). The acetone exchange method was utilized to measure ^2^H enrichment in plasma. Briefly, 5 ul of plasma was transferred into the inlet of autosampler vials followed by adding 2 ul of 10N NaOH and 5 ul of acetone. Following 4 hrs of incubation at room temperature, enrichment of analytes with head space injection was determined using mass to charge ratio (m/z) of 58 (M + 0) and 59 (M + 1) for acetone shown at ~1.2 min. For the analysis of alanine enrichment, the myofibrillar and mitochondrial fractions were hydrolyzed into amino acids at 100 °C for 16 to 24 h. Then, free amino acids from the hydrolyzed sample were extracted via cation-exchange resin columns and dried under Speed Vac (Savant Instruments, Farmingdale, NY, USA). ^2^H enrichment of alanine was determined on the tert-butyldimethylsilyl derivative and ions of m/z of 260 (M + 0) and 261 (M + 1) were monitored. 

### 2.9. Calculations of Muscle Protein Kinetics

The calculation of muscle protein fractional synthesis rate (FSR, %/time) was predicated on precursor-product rule [35]. Briefly, FSR was calculated as changes in product enrichment (i.e., ^2^H enrichment of protein bound alanine in muscle) divided by steady state enrichment of precursor (i.e., free alanine), multiplied by 100. The precursor enrichment was inferred from the ^2^H enrichment of body fluid (e.g., plasma) accounting for ~3.7 potential ^2^H exchange sites of alanine [36].
FSR (%/t) = [(E_Ala_)/(E_BW_ × 3.7)] × 100(1)
*k*_s_ = −ln (1 − FSR)/t (2)
Absolute synthesis (mg/day × 28 days) = *k*_s_ (day^−1^) × muscle pool size (mg) × 28 days (3)
where enrichment (E) is expressed as mole percent excess (MPE) calculated as tracer to tracee ratio (TTR)/(TTR + 1); t is time of labeling; and ks represents the fractional synthesis rate constant [37]. The absolute synthesis rate of muscle protein was calculated by multiplying the cumulative ks over 28 days of the labeling period by muscle protein pool size with the assumption of 12% of total muscle wet weight, which is myofibrillar protein in muscle [38].

### 2.10. RNA Isolation, cDNA Synthesis, and Quantitative Real-Time PCR

Gastrocnemius muscle, 15–20 μg, was homogenized into a 0.5 mL TRIzol reagent (Invitrogen, Carlsbad, CA, USA) and RNA was isolated according to the manufacturer’s instruction. Samples were treated with DNase, chloroform extract, and re-suspended in 30 µL RAase-free-water. Isolated RNA purity and concentration were confirmed using Nanodrop2000 spectrometer (Thermoscientific, Wilmington, NC, USA). Then, RNA was reverse transcribed into cDNA using TOPscriptTM RT Drymix (RT200, Enzynomics, Daejon, Korea). Quantitative polymerase chain reaction PCR were performed using a TOPreal™ qPCR 2× premix (SYBR Green) kit (RT620US, Enzynomics, Daejon, Korea). The following primers were used for RT-PCR assessment (Table 2): GAPDH, MyoD, Myogenin, MuRF1, Atrogin-1, LC3, p62, MuSK, and AchR, and GAPDH. Final quantification of gene expression was calculated using the ΔΔCT method. Relative quantification was calculated as 2−ΔΔCT.

### 2.11. Western Blot Analysis

Gastrocnemius muscle was homogenized using lysis buffer (#9803S, Cell signaling Technology, Beverly, MA, USA) containing a protease inhibitor cocktail (5871S, Cell signaling Technology, MA, USA) and phosphatase inhibitor cocktail (P5726S, Sigma-Aldrich, St Louis, MO, USA). Homogenates were centrifuged at 4 °C at 12,000 rpm for 15 min, and total protein concentration of the supernatant was determined using Bradford method [39]. The samples were boiled at 95 °C for 5 min. For each sample, a total protein concentration of 15 µg was resolved by SDS-PAGE using 10% gels and transferred to polyvinylidene fluoride (PVDF) membranes at 70 volts for 90 min. After blocking the membrane for 1 h at room temperature, primary antibodies, including Akt1 (#9272), p-Akt1 (Ser473) (#9271), mTOR (#2983), p-mTOR (Ser2448) (#2971), p70s6k (#9202), p-p70s6k (Ser371) (#9208), COX IV (#4844S), and GAPDH (#2971, Cell signaling technology, MA, USA), were incubated overnight at 4 °C. The membrane was then washed with TBS-T and incubated with Anti-rabbit IgG (HRP) (#7074, Cell signaling technology, MA, USA) at room temperature for 1 hr. Immunodetection was carried out with ECL detection reagent (GE healthcare, Buckingham, UK). Protein amount was analyzed using image J software (National Institutes of Health).

### 2.12. Statictical Analysis

Normality of distribution was assessed by the Shapiro–Wilks test. One-way analysis of variance (ANOVA) was performed to determine the main effects of intervention groups on variables, and Fisher’s least significant difference (LSD) post hoc test was used for multiple comparisons. Since some of PCR data were not normally distributed, Kruskal Wallis test was performed. Two-way ANOVA was performed to determine the main effects of ACV, EAA, and RT and interaction of EAA and RT among four ACV groups, and LSD post hoc test was used when two-way ANOVA revealed significances. Statistical analyses were performed using SPSS for window version 23.0 (SPSS, Inc., Chicago, IL, USA). Statistical significance was set at *p* < 0.05. All data were expressed as mean ± SE.

## 3. Results

### 3.1. Body Composition and Mucle Function

We found that hindlimb muscle mass (including muscles of soleus, plantaris, gastrocnemius, extensor digitorum longus, tibialis anterior, and quadriceps femoris) was significantly increased in all ACV treatment groups compared to CON, particularly in RT groups (i.e., ACV+RT and ACV+RT+EAA) compared to non-RT groups (Figure 1B). In agreement with the total hindlimb muscle mass, there was a significant increase in each hindlimb muscle mass in both RT groups compared to non-RT groups (Figure 1C–H). To determine the muscle function and quality, we measured the four-limb grip strength in all groups and changes in maximal carrying load in ACV+RT and ACV+RT+EAA. The maximal grip strength was significantly higher in all ACV groups compared with CON, and among ACV groups the ACV+EAA+RT had the highest maximal grip strength (Figure 1I). In the RT groups, changes in maximal carrying load normalized by body weight were significantly higher in ACV+RT+EAA compared to ACV+RT over 4 weeks (Figure 1J). ACV+EAA and ACV+RT+EAA increased the grip strength compared to CON and EAA groups (i.e., ACV+EAA and ACV+RT+EAA) induced greater grip strength than non-EAA groups in the maximal force normalized to hindlimb muscle mass (Figure 1K). 

### 3.2. Myofibrillar Protein Synthesis Rate

To explore the changes in muscle mass and strength in response to respective treatments, we measured the myofibrillar protein synthesis rate over 4 weeks. As a result, ACV, ACV+RT, and ACR+RT+EAA significantly increased myofibrillar protein synthesis rates in gastrocnemius (Figure 2A). All ACV treatment groups increased myofibrillar protein synthesis rates in tibialis anterior compared to CON (Figure 2B). Interestingly, the ACV+EAA+RT had a significantly higher protein synthesis rate in the soleus muscle than other ACV groups (Figure 2C).

### 3.3. Protein Turnover Signaling Pathway

Regarding protein synthesis, there was no difference between groups in Akt1, mTORC1, and p70s6k (Figure 3A–C), but with respect to myogenesis, there was a significant increase in MyoD mRNA expression in ACV+EAA, ACV+RT, and ACV+EAA+RT compared to CON (Figure 3D). Regarding protein breakdown, the mRNA expression of Atrogin-1 but not of MuRF1 in UPS was significantly decreased in the ACV+RT, ACV+EAA, and ACV+EAA+RT compared to CON (Figure 3E). However, there were no significant differences between groups in mRNA expressions of LC3 and p62, markers for autophagy (Figure 3F). Interestingly, there was a positive linear correlation between hindlimb muscle mass and MyoD mRNA expression (Figure 3G).

### 3.4. Mitochondrial Protein Kinetics and Neuromuscular Junction Stability

To determine if mitochondria and neuromuscular junction play a role in EAA-mediated improvements in muscle quality, we analyzed a marker for mitochondrial protein content (cytochrome c oxidase, COX IV) [40,41], mitochondrial protein turnover rate, and neuromuscular junction (NMJ) stability-related mRNA expression in the gastrocnemius. First, we found that COX IV contents were not different among all groups (Figure 4A). However, we found that mitochondrial protein synthesis was significantly increased in ACV+RT and ACV+RT+EAA compared to CON (Figure 4B). The mRNA expression of MuSK, a marker of NMJ stability, was significantly higher in ACV+RT+EAA than CON (Figure 4C). There was no positive relationship between MuSK mRNA expression and muscle quality when maximal carrying load was normalized to muscle mass (Figure 4D). However, we found a positive correlation between MuSK mRNA expression and muscle strength when 1 maximal carrying load was normalized to body weight (Figure 4E).

## 4. Discussion

To our knowledge, this is the first study to demonstrate that: (1) RT synergistically increases muscle mass with the treatment of soluble form of the activin type IIB receptor (sActRIIB-Fc, ACV); and (2) dietary EAAs increase the quality (i.e., strength/muscle mass) of muscle hypertrophied with either treatment of ACV or ACV+RT in mice.

In previous animal studies, various forms of myostatin inhibition have shown to significantly increase muscle mass in pathophysiologic (i.e., cancer cachexia, Duchenne muscular dystrophy and sarcopenia) and normal mice models [6,15,42]. Among others, ACV has been studied extensively. For example, Latres et al. reported that TA muscle weight was increased by approximately 49% with ACV administrations twice per week for 3 weeks [17]. In agreement with the previous study, we found that ACV alone significantly increased muscle mass (more than 13% in hindlimb muscles) as compared to CON. It is well-known that RT and EAAs are physiologically potent anabolic stimuli for muscle accretion through stimulation of protein synthesis and/or suppression of protein breakdown in both humans [25] and mice [43]. In concert, we found that RT increased hindlimb muscle mass by 28% compared to CON, which represents about 14% of additional improvement compared to ACV alone. Similar to the previous finding [44], we observed that consumption of additional dietary EAAs beyond the normal dietary intake did not result in muscle accretion above ACV or ACV+RT. Results from mouse study are contrary to previous human findings [45], in which dietary EAA ingestions over weeks improved lean body mass. The discrepancy between mice and humans might be because the amount of EAAs provided in mice was not sufficient (~0.06% of daily calorie intake) compared to that of humans (~5% of daily calorie intake).

Muscle mass is the direct result of balance between rates of muscle protein synthesis and breakdown over time. In the current study, integrated muscle protein synthesis was similarly increased in all ACV-treated groups compared to CON despite the fact that muscle gains were greater with the addition of RT (i.e., ACV+RT and ACV+RT+EAA) compared to non-RT groups (i.e., ACV and ACV+EAA). Thus, the results imply that the differences in changes in muscle mass among groups were largely due to differences in integrated MPB. Myostatin acts largely on stimulation of MPB [46]. Thus, inhibition of myostatin may attenuate MPB, which in turn reduces intramyocellular AA availability (as MPB is the largest source of the availability) and thus negatively affect the potential of MPS [21], which might however be compensated for by another stimulus for MPS (i.e., RT) [47]. Taken together, the gains in muscle mass by myostatin inhibition was further amplified by RT due largely to reductions in MPB.

Muscle protein turnover (i.e., MPS and MPB) is regulated by (in)activation of implicated signaling pathways. Protein synthesis is mainly up-regulated by activation of Akt1/mTORC1 axis through acting on protein translation [48] and myogenesis, both of which are known to be activated by RT or EAAs [25] but inhibited by myostatin [16]. On the other hand, protein breakdown is largely up-regulated by the activation of UPS [49]. Contrary to the notion, we found no changes in activations of Akt1/mTORC1 pathways among treatments in the current study. The absence of changes in the implicated signaling pathway for protein synthesis is not uncommon. For example, it was shown that muscle mass was increased without activation of Akt1/mTORC1 signaling after 4 weeks of RT [50] or 10 weeks of ACV treatment [51]. On the other hand, studies have shown that 8 weeks of RT [42] or 12 weeks of ACV treatment [52] increased Akt1 activity. Another possibility for the discrepancy may include differences in study design including the timing of tissue harvest. For example, the timing of muscle tissue collection complicates the interpretation of signaling data due to the transient nature of Akt1/mTORC1 activation [53,54].

In agreement with the previous studies in older humans after 12 weeks of RT [55] and in vitro with treatment of leucine [52], we found that the addition of EAAs and/or RT increased mRNA expression of MyoD, a marker of myogenesis (i.e., muscle formation and muscle growth). In terms of UPS, we found that atrogin-1, but not MuRF1, mRNA expression was significantly suppressed with the addition of EAAs or RT compared to CON. Taken together, mechanisms responsible for differences in muscle mass gains among treatments above ACV alone are unclear but may be explained at least in part by changes in myogenin and atrogin-1 mRNA expression.

Muscle mass is the main determinant of muscle strength [56]. In the current study, we also found that improvements in muscle strength was in proportional to gains in muscle mass in general. Interestingly, addition of dietary EAAs (i.e., ACV+EAA, ACV+RT+EAA) led to a greater muscle strength, compared to ACV or RT+ACV, respectively, without further muscle mass gains. This implies that dietary EAAs improve muscle quality (strength/mass). We postulated that this improvement resulting from EAA consumption is due in part to up-regulation of protein turnover in contractile and/or mitochondrial proteins. In accordance with this notion, previous studies reported that EAA supplementation induced a greater strength improvement despite no changes in muscle mass [45], which was accompanied with increased mitochondrial biogenesis [28]. In addition, combination of BCAA ingestion with acute resistance exercise induced a greater muscle myofibrillar fractional synthesis rate up to 22% than resistance exercise alone [57]. However, we found that addition of dietary EAAs did not increase rates of mitochondrial protein synthesis, compared to ACV or RT. Instead, we found that addition of RT to ACV (i.e., ACV+RT or ACV+RT+EAA) induced greater rates of mitochondrial protein synthesis than CON. These results indicate the existence of other mechanism(s) for the EAA-mediated increase in muscle quality. Alternatively, Chevessier et al. reported that mutation of MuSK gene, an important component for NMJ stability which is closely related to muscle function and formation, inhibited muscle strength gains and reduced specific force (i.e., muscle quality) in young mice [58]. Thus, we investigated if changes in NMJ stability play a role for the EAA-induced improvement of muscle quality or muscle force generation normalized to muscle mass (i.e., maximal carrying load/muscle mass or grip strength/muscle mass). In the current study, we did not find any positive effect of dietary EAA on MuSK mRNA expression and no correlation between muscle quality and MuSK mRNA expression, indicating no role of NMJ stability for the EAA-mediated improvement in muscle quality. Alternatively, it is possible that dietary EAAs alter dynamics of mitochondrial fusion and fission that may affect ATP production [59] and in turn skeletal muscle contractility [60]. Thus, further studies are required to elucidate the mechanism(s) by which dietary EAAs induce improvements in muscle quality.

There are several potential limitations in the present study. First, the timing of tissue harvest might have masked the effect on molecular signaling implicated in protein turnover. In the present study, the muscle tissues were harvested 4 days (after the last RT session) [54] or 11 days (after the last ACV injection) after the last treatment. Thus, it is possible that (in)activated molecular signaling following respective stimuli might return to the baseline. Second, the null effect of dietary EAAs on muscle (protein) gains may be due to the fact that the amount of EAAs supplemented (although it was the maximum amount that can be delivered orally) was not sufficient as the caloric contents of the supplemented EAAs were less than 1% of total daily. Furthermore, the dosage (i.e., 1.5g/kg/day for mice) was approximately six times more than the RDA for total EAAs for humans, which is approximately 0.35 g EAAs/kg/day, although requirement of EAAs for mice may be different from that of humans. Third, the current study design does not elucidate which components (e.g., BCAAs) of EAAs causes improvements in muscle quality. However, it is clear that all of EAA components are required for improvements of both muscle mass [56,61,62] and quality. Future investigation into elucidation of major component(s) of EAAs and/or molecular mechanisms underlying the muscle quality-enhancing effect of dietary EAAs (or its components) are warranted. Lastly, a small number of sample size might mask the effect of treatments (particularly, dietary EAAs) in muscle mass and protein synthesis rate.

## 5. Conclusions

In the present study, we showed that addition of resistance exercise training, but not dietary EAAs, to the myostatin inhibition further increased muscle mass through the attenuation of muscle protein breakdown with proportionate improvements in muscle strength. Interestingly, addition of dietary EAAs to the myostatin inhibition with or without resistance exercise training improved muscle quality. Thus, dissection of the underlying mechanism(s) behind the combined positive effect of dietary EAAs and resistance exercise training on muscle mass and quality can shed light on the discovery of effective therapeutics against muscle wasting such as sarcopenia.

## Figures and Tables

**Figure 1 nutrients-13-01508-f001:**
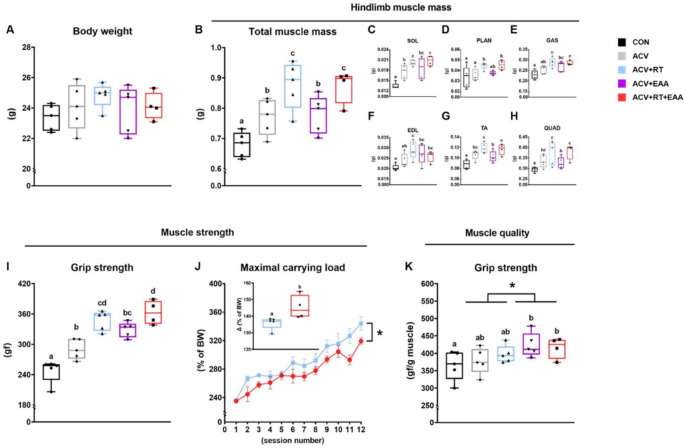
Dietary essential amino acids (EAA) improved quality of the hypertrophied muscle by RT and/or myostatin inhibition. (**A**) body weight. (**B**) Total hindlimb muscle mass (soleus, plantaris, gastrocnemius, extensor digitorum longus, tibialis anterior and quadriceps femoris). (**C**–**H**) Individual hindlimb muscle mass. (**I**) Grip strength. (**J**) Changes in maximal carrying load over 4 weeks of resistance exercise training (RT) and maximal carrying load at the completion of RT (inset) in both RT groups (i.e., ACV (myostatin inhibitor) + EAA and ACV+ EAA+ RT). (**K**) Muscle quality (grip strength normalized by hindlimb muscle mass). Data are presented as mean ± S.E. ^a,b,c,d^ Groups not sharing the same letter are significantly different (*p* < 0.05), * *p* < 0.05, main effect for EAA. BW, body weight. SOL, soleus; PLAN, plantaris; GAS, gastrocnemius; EDL, extensor digitorum longus; TA, tibialis anterior; QUAD, quadriceps femoris.

**Figure 2 nutrients-13-01508-f002:**
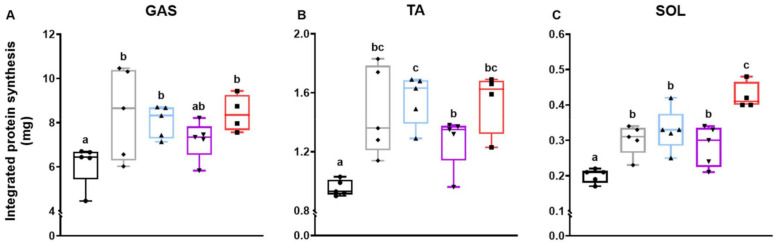
Integrated myofibrillar protein synthesis rate over 4 weeks of respective treatments. Absolute protein synthesis in (**A**) gastrocnemius, (**B**) tibialis anterior, and (**C**) soleus. Integrated protein synthesis rate was calculated as the product of 28-day cumulated muscle FSR and muscle protein pool size. Data are presented as mean ± S.E. ^a,b,c^ Groups not sharing the same letter are significantly different (*p* < 0.05). GAS, gastrocnemius; TA, tibialis anterior; SOL, soleus.

**Figure 3 nutrients-13-01508-f003:**
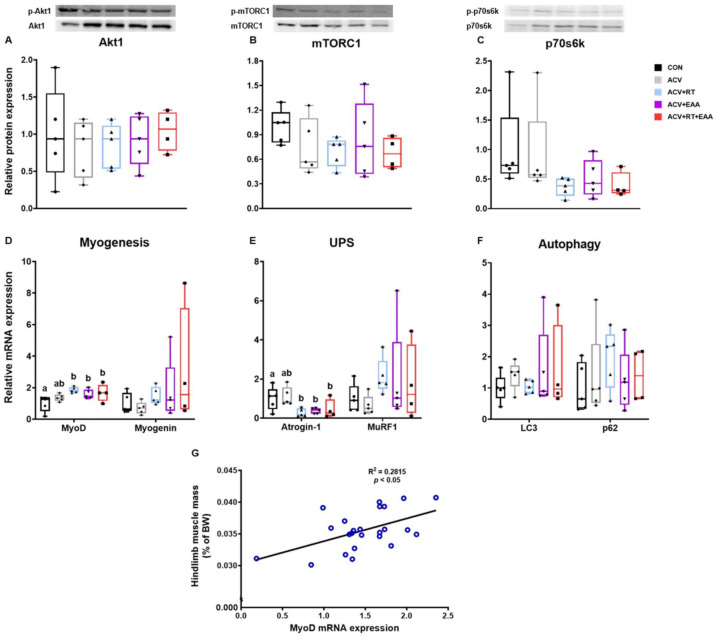
Gains in muscle mass with treatment of EAA and/or RT were associated with increase in MyoD and decrease in Atrogin-1 mRNA expression in gastrocnemius muscle. (**A**–**C**) Relative protein expression of Akt1, mTORC1, and p70s6k. (each protein was normalized to GAPDH). (**D**–**F**) Relative mRNA expression of myogenesis-related genes and ubiquitin proteasome-related genes and autophagy-related genes (expression levels were normalized to GAPDH). (**G**) Correlation between MyoD mRNA expression and muscle mass. Data are presented as mean ± S.E. ^a,b^ Groups not sharing the same letter are significantly different (*p* < 0.05). Akt1, protein kinase B; mTORC1, mammalian target of rapamycin complex 1; p70s6k1, ribosomal protein S6 kinase beta-1; MyoD, myoblast determination protein 1; Myogenin, myogenic factor 4; UPS, ubiquitin proteasome system; Atrogin-1, muscle atrophy F-box protein; MuRF1, muscle ring finger protein-1; LC3, microtubule associated protein 1A/1B light chain 3; p62, ubiquitin-binding protein p62; BW, body weight.

**Figure 4 nutrients-13-01508-f004:**
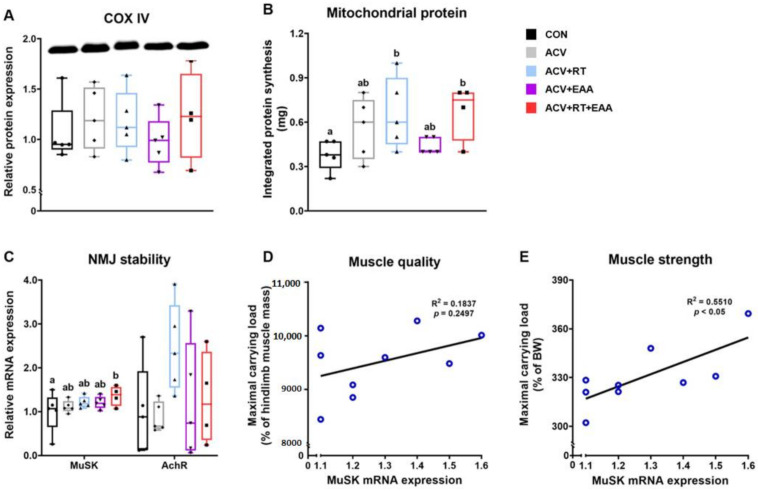
EAA-mediated improvements in muscle quality were mediated by other than changes in mitochondrial protein turnover or neuromuscular junction stability. (**A**) Relative protein expression of COX IV (COX IV was normalized to GAPDH). (**B**) Integrated mitochondrial protein synthesis over 28 days in gastrocnemius. (**C**) Relative mRNA expression of NMJ stability-related genes MuSK and AchR in gastrocnemius. (**D**,**E**) Correlations between MuSK mRNA expression and muscle quality (i.e., maximal carrying load normalized to muscle mass) and muscle strength (i.e, maximal carrying load normalized to body weight). Data are presented as mean ± S.E. ^a,b^ Groups not sharing the same letter are significantly different (*p* < 0.05). COX IV, cytochrome c oxidase subunit 4; NMJ, neuromuscular junction; MuSK, muscle-specific tyrosine kinase; AchR, acetylcholine receptor; BW, body weight.

**Table 1 nutrients-13-01508-t001:** Composition and relative intake of EAAs.

Amino Acid	Percentage (%)	Dietary Intake (g/kg/day)
Histidine	10	0.150
Isoleucine	10	0.150
Leucine	21	0.315
Lysine	18	0.270
Methionine	4	0.060
Phenylalanine	12	0.180
Threonine	14	0.210
Valine	10	0.150
Tryptophan	1	0.015
Total	100	1.5

**Table 2 nutrients-13-01508-t002:** The list of primers sequences.

No.	Genes		Sequence
1	MyoD	Forward	5′ ACCAACGCTGATCGCCGCAA 3′
		Reverse	3′ GCAGCGGTCCAGGTGCGTAG 5′
2	Myogenin	Forward	5′ TGTGTCGGTGGACCGGAGGA 3′
		Reverse	3′ CCGCTGGTTGGGGTGGAGCA 5′
3	Atrogin-1	Forward	5′ GACTGGACTTCTCGACTGCC 3′
		Reverse	3′ TCAGGGATGTGAGCTGTGAC 5′
4	MuRF1	Forward	5′ AAGCAGGTGCCACTCTCTGT 3′
		Reverse	3′ AGCTTCACACCTGTCCTTCG 5′
5	LC3	Forward	5′ CACTGCTCTGTCTTGTGTAGGTTG 3′
		Reverse	3′ TCGTTGTGCCTTTATTAGTGCATC 5′
6	P62	Forward	5′ CCCAGTGTCTTGGCATTCTT 3′
		Reverse	3′ AGGGAAAGCAGAGGAAGCTC 5′
7	MuSK	Forward	5′ TGAGAACTGCCCCTTGGAACT 3′
		Reverse	3′ GGGTCTATCAGCAGGCAGCTT 5′
8	AchR	Forward	5′ CATCGAGGGCGTGAAGTACA 3′
		Reverse	3′ ATTCCTCAGCGGCGTTATTG 5′
9	GAPDH	Forward	5′ CACCATCTTCCAGGAGCGAG 3′
		Reverse	3′ CCTTCTCCATGGTGGTGAAGAC 5′

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
