# Peer review of "Myostatin Inhibition-Induced Increase in Muscle Mass and Strength Was Amplified by Resistance Exercise Training, and Dietary Essential Amino Acids Improved Muscle Quality in Mice"

_nutrients, 2021, doi:10.3390/nu13051508_

Round 1

Reviewer 1 Report

In this manuscript by Jang J.et al., the authors negatively modulate myostatin dependent pathways in mice to trigger a shift of the balance between protein synthesis and degradation, thus stimulating muscle growth and force. To stimulate these responses the authors use also a nutritional supplementation (EAA) based on amino-acids and a program od resistance training (RT) or a combination of both. The topic is of relevant interest, since strategies to counteract the progressive muscle loss concomitant to cancer cachexia and sarcopenia, associated to ageing or chronic comorbidities, are urgently needed, although several limitations have been found for the translation to humans. The manuscript is original however a major revision is needed.

Some points should be clarified.

Among the mechanisms of action of myostatin there is also a negative effect on satellite cells activation/proliferation (McCroskery et al., 2003). The inhibition of myostatin have been shown to favor satellite cells activation and proliferation. This information is missing in the introduction, but this could be another mechanism to take into consideration.

The authors used seven-weeks old mice, that are pretty young mice, thus it’s not clear  if the interventions would have the same results in older mice. Indees most of the conditions leading to sarcopenia in humans develop in the middle/advanced age. Maybe using 12 or 18 months old mice could have been more meaningful. Should justify their choice and implications.

About the experimental design: how can the author exclude that the results obtained with RT or RT+EAA are due to the ACV treatment and not to the RT or RT+EAA alone?? maybe a control is missing in the study.

How did the author estimate the sample size? Are five animals per group enough? The results for the ACV+RT+EAA group always report only 4 points. The authors should justify why there is a missing data point.

The authors report the muscle mass as a sum of  the mass of  main hindlimb muscles (it should be clear also from the indication on the Y axis of Fig 1B. Then they normalize the grip strength by muscle mass: are the hindlimbs mainly involved in the strength test?

About the mitochondrial protein kinetics (Fig. 4A and 4B): the authors show the expression of COX IV protein (Fig 4A), however it’s not clear if this marker could represent  the amount of mitochondria at 28 days (??). Maybe a comparison/subtraction between a group at baseline and the endpoint could be more informative. Again it is not clear how the authors evaluated the mitochondrial integrated protein synthesis: did they isolate mitochondria. There is just a time point shown and so I’d find it difficult to call this a “kinetic”.

Figure 3 Check the caption title (a word seem to be missing).

Raw 150-151: predicated à predicted.

Author Response

We appreciate your time and effort in providing these valuable comments.

REVIEWER #1: In this manuscript by Jang J.et al., the authors negatively modulate myostatin dependent pathways in mice to trigger a shift of the balance between protein synthesis and degradation, thus stimulating muscle growth and force. To stimulate these responses the authors use also a nutritional supplementation (EAA) based on amino-acids and a program of resistance training (RT) or a combination of both. The topic is of relevant interest, since strategies to counteract the progressive muscle loss concomitant to cancer cachexia and sarcopenia, associated to ageing or chronic comorbidities, are urgently needed, although several limitations have been found for the translation to humans. The manuscript is original however a major revision is needed.

COMMENT #1: Among the mechanisms of action of myostatin there is also a negative effect on satellite cells activation/proliferation (McCroskery et al., 2003). The inhibition of myostatin have been shown to favor satellite cells activation and proliferation. This information is missing in the introduction, but this could be another mechanism to take into consideration.

RESPONSE #1: As suggested, it has been added (Line 63-64).

COMMENT #2: The authors used seven-weeks old mice, that are pretty young mice, thus it’s not clear if the interventions would have the same results in older mice. Indeed most of the conditions leading to sarcopenia in humans develop in the middle/advanced age. Maybe using 12- or 18-months old mice could have been more meaningful. Should justify their choice and implications.

 RESPONSE #2: We agree with your point that myostatin inhibition study using 12- or 18-months old mice could be more meaningful. However, the study intention was not specifically on aging muscle but on loss of muscle in general. As you suggested, however, we will be conducting studies specifically focusing on aging muscle in the future research.

COMMENT #3: About the experimental design: how can the author exclude that the results obtained with RT or RT+EAA are due to the ACV treatment and not to the RT or RT+EAA alone?? maybe a control is missing in the study.

RESPONSE #3: The primary hypothesis of the current study is to examine if addition of RT and/or EAAs improves muscle mass and strength compared to ACV alone. Moreover, the effects of RT or RT+EAA alone investigated in the previous studies have been well established. In the current study, we included CON or ACV groups which act as baseline comparing to other groups. Thus, we believe that controls to examine the effect of RT or RT+EAA are included.

COMMENT #4: How did the author estimate the sample size? Are five animals per group enough? The results for the ACV+RT+EAA group always report only 4 points. The authors should justify why there is a missing data point.

RESPONSE #4: The sample size was calculated using G*power software version 3.1.9.7 (Heinrich Heine University, Düsseldorf, Germany). The sample size was chosen as 25 (5 per group) in order to detect a minimum difference between 5 groups (α = 0.05, β = 0.80) (Line 115-118). We administered EAA or BCAA orally for the duration of the experiment. Among them, one mouse in the ACV+RT+EAA group died in the middle of the experiment so that the data with the mouse were excluded (Line 115).

COMMENT #5: The authors report the muscle mass as a sum of the mass of main hindlimb muscles (it should be clear also from the indication on the Y axis of Fig 1B. Then they normalize the grip strength by muscle mass: are the hindlimbs mainly involved in the strength test?

RESPONSE #5: It has been adjusted. (Fig 1A). As we conducted a whole limb grip strength test which requires the involvement of hindlimb muscle strength to a large extent, we believe that the normalization using the hindlimb muscle mass is reasonable. Moreover, the normalization using body weight shown in many previous studies is less reliable as other issues such as adipose, liver, kidney and etc. are not involved in generation of grip strength.  

COMMENT #6: About the mitochondrial protein kinetics (Fig. 4A and 4B): the authors show the expression of COX IV protein (Fig 4A), however it’s not clear if this marker could represent the amount of mitochondria at 28 days (??). Maybe a comparison/subtraction between a group at baseline and the endpoint could be more informative. Again it is not clear how the authors evaluated the mitochondrial integrated protein synthesis: did they isolate mitochondria. There is just a time point shown and so I’d find it difficult to call this a “kinetic”.

RESPONSE #6: First, the expression of COX IV is commonly used as a surrogate marker for mitochondrial content. And because it is not possible to obtain muscle pre and post-intervention in mouse study in contrast to the case of human study, having a control group as in the current study is sufficient. Second, we isolated muscle mitochondrial fraction using an established protocol (please see below) (Line 151-157) and then measure magnitude of labelling by 2H2O over 28 days. The integrated mitochondrial protein synthesis rate was normalized to COX 4 protein for group comparisons.

Reference: Mitochondrial isolation using the protocol - Liao, P.C.; Bergamini, C.; Fato, R.; Pon, L.A.; Pallotti, F. Isolation of mitochondria from cells and tissues

COMMENT #7: Figure 3 Check the caption title (a word seem to be missing).

RESPONSE #7: It has been adjusted (Line 279-280).

COMMENT #8: Raw 150-151: predicated a predicted.

RESPONSE #8: We used “predicated” as the point of the sentence is that the calculation was performed, based on (or predicated on) the precursor-product rule.

Reviewer 2 Report

In the manuscript entitled “Myostatin inhibition-induced increase in muscle mass and strength was amplified by resistance exercise training, and dietary essential amino acids improved muscle quality in mice” by Jan et al, the authors addressed the hypothesis of whether resistance training increases the anabolic potential of myostatin inhibition and whether EAAs enhance muscle quality through stimulations of mitochondrial biogenesis and/or muscle protein turnover. Although the Authors provide several experimental approaches to answer the hypothesis, it is at the current stage no fully answered. For instance, the Authors state that “EAAs enhance muscle quality through stimulations of mitochondrial biogenesis”. There is no data provided in the manuscript answering this part of the hypothesis. Instead, the Authors evaluated whether “mitochondria and neuromuscular junction play a role in EAA-mediated improvements in muscle quality”. This and other issues detailed below must be addressed by the Authors before a final decision.

Comments

  • Introduction: The authors should include in the introduction the basis to study the neuromuscular junction in the context of the proposed study. Figure 4 presents data for neuromuscular junction, but it is not clear the reason to include it here because the data is concentrated in the muscle function itself and not how the nerve activation would influence or be influenced by the treatments proposed here. The Authors should also include an explanation of the terms Muscle Quality and Muscle Strenght and how these parameters are calculated and their meaning. It is not clear the questions that these parameters can answer in the current manuscript.

  • Line 126: The grip strength test is not a reliable test to measure force because the fact to pull the tail of the mouse induces stress and pain. Therefore, it is not possible to know whether the mouse is releasing the bar because of fear, pain or because it can't hold it any longer. The authors should evaluate the muscle force capacity using an in vitro setup or calculate the work done from the ladder.

  • Line 213: The Authors should clarify the meaning of 1RM and how it was measured.

  • Figures: Since the Authors already show the value of each data point, the Authors should change the format of the data display from bars to box plot. The box plot format will display better the distribution of the data. Moreover, due to the large scale, it is difficult to appreciate the differences between groups.

  • Figure 1: The Authors should include the meaning of the letters a, b, c, and d.

  • Line 229: The Authors should clarify how the samples for the myofibrillar protein synthesis rate were taken, including a description of the method of anesthesia, probe sampling, and care of the animal after the procedure. In case that the samples were taken after had sacrificed the mice, then the Authors should change the word "over' to after. The meaning of over also induces the idea of continuing sampling during the period of 4 weeks.

  • Figure 2: To be able to have a better comparison to other publications and thus further understand the meaning of the parameter integrated protein synthesis, the Authors should include the total muscle weight of all three muscles.

  • Figure 3: The Authors should run and show the results of an outlier test on the PCR data to check whether these points beyond the distribution of the data can be considered in the analysis.

  • Figure 3: The authors should clarify to whom the data was normalized. Moreover, the Authors should show more representative immunoblotting due to the discrepancy between the summarized data and the representative immunoblotting. For instance, it is hard to see that in controls mTORC1 and p-mTOR have the same intensity.

  • Figure 4: The Authors should clarify what kind of protein is COX IV, why it was selected to analyze mitochondrial protein content, and what was the housekeeping protein used for normalization.

  • Figure 4: The Authors should provide information regarding the muscle fiber type before and right after the treatments and muscle fiber diameter. These parameters are classical and relevant to understand the changes induced by the treatments applied here.

  • Discussion: The Authors should discuss two papers related to the current manuscript:

Effect of mechanistic/mammalian target of rapamycin complex 1 on mitochondrial dynamics during skeletal muscle hypertrophy. Uemichi K, Shirai T, Hanakita H, Takemasa T. Physiol Rep. 2021 Mar;9(5):e14789. doi: 10.14814/phy2.14789.

Pretreatment with a soluble activin type IIB receptor/Fc fusion protein improves hypoxia-induced muscle dysfunction. Pistilli EE, Bogdanovich S, Mosqueira M, Lachey J, Seehra J, Khurana TS. Am J Physiol Regul Integr Comp Physiol. 2010 Jan;298(1):R96-R103. doi: 10.1152/ajpregu.00138.2009.

High-fat diet suppresses the positive effect of creatine supplementation on skeletal muscle function by reducing protein expression of IGF-PI3K-AKT-mTOR pathway. Ferretti R, Moura EG, Dos Santos VC, Caldeira EJ, Conte M, Matsumura CY, Pertille A, Mosqueira M. PLoS One. 2018 Oct 4;13(10):e0199728. doi: 10.1371/journal.pone.0199728. eCollection 2018.

  • Line 356: The Authors should explain better the meaning of correlating the 1RT/muscle mass with MuSK expression to give the concept of muscle quality. It is hard to grasp the meaning of this data when the percentage of 1RT/ muscle mass is on the scale of thousands of percent or 1RT/ body mass over hundreds of percent. The physiological question behind this is how the muscle quality (and which quality the Authors want to emphasize with these parameters) is measured with these correlations. Moreover, the Authors should also provide the slope of the correlation to understand the gain of the muscle quality due to the treatment.

  • Line 361: As the Authors correctly pointed out one of the major limitations of the study, where the muscles were harvested 4 and 11 days after the RT and ACV treatments, respectively. As the Authors also correctly discussed, this large time gap between treatment and tissue harvesting substantially affected the data and thus the interpretation of the results. Therefore, it is necessary then to provide data just after the last treatment to understand the effect of the treatments and comprehend better the physiological effect of such treatment.

Another important point discussed by the Authors is the limited number of animals used per study. Whenever possible, the Authors should include another set of animals to see if the Authors can replicate their own data.

  • Line 372: the authors should decide whether use branched-chain amino acid or the abbreviation BCAA.

Author Response

We appreciate your time and effort in providing these valuable comments.

REVIEWER #2: In the manuscript entitled “Myostatin inhibition-induced increase in muscle mass and strength was amplified by resistance exercise training, and dietary essential amino acids improved muscle quality in mice” by Jan et al, the authors addressed the hypothesis of whether resistance training increases the anabolic potential of myostatin inhibition and whether EAAs enhance muscle quality through stimulations of mitochondrial biogenesis and/or muscle protein turnover. Although the Authors provide several experimental approaches to answer the hypothesis, it is at the current stage no fully answered. For instance, the Authors state that “EAAs enhance muscle quality through stimulations of mitochondrial biogenesis”. There is no data provided in the manuscript answering this part of the hypothesis. Instead, the Authors evaluated whether “mitochondria and neuromuscular junction play a role in EAA-mediated improvements in muscle quality”. This and other issues detailed below must be addressed by the Authors before a final decision.

COMMENT #1: Introduction: The authors should include in the introduction the basis to study the neuromuscular junction in the context of the proposed study. Figure 4 presents data for neuromuscular junction, but it is not clear the reason to include it here because the data is concentrated in the muscle function itself and not how the nerve activation would influence or be influenced by the treatments proposed here. The Authors should also include an explanation of the terms Muscle Quality and Muscle Strength and how these parameters are calculated and their meaning. It is not clear the questions that these parameters can answer in the current manuscript.

RESPONSE #1: We appreciate your attentive comment. It has been added (Line 93-95). The meaning of muscle quality was added (Line 69-70, 385-386).

COMMENT #2: Line 126: The grip strength test is not a reliable test to measure force because the fact to pull the tail of the mouse induces stress and pain. Therefore, it is not possible to know whether the mouse is releasing the bar because of fear, pain or because it can't hold it any longer. The authors should evaluate the muscle force capacity using an in vitro setup or calculate the work done from the ladder.

RESPONSE #2: We appreciate your comment. We are aware of the point so that there was the period of familiarization to ensure that mice are comfortable with the grip strength test and the conductor to perform the test is particularly assigned without randomly changing conductors in order to enhance reliability. However, due to the experimental condition, we were unable to evaluate the muscle strength using an in vitro setting, and as another alternative to the grip strength test, the 1RM measured through the ladder climbing exercise was presented in Figure 1. J in RT groups as a result representing whole-body muscle strength.  

COMMENT #3: Line 213: The Authors should clarify the meaning of 1RM and how it was measured.

RESPONSE #3: Due to the potential confusion on the term 1RM, we replaced it with maximal carrying load, defined as the maximal weight that mice successfully carried to the top of the ladder. The load carried in the last successful climbing was considered as the maximal carrying load at the exercise session, and was not measured separately.

COMMENT #4: Figures: Since the Authors already show the value of each data point, the Authors should change the format of the data display from bars to box plot. The box plot format will display better the distribution of the data. Moreover, due to the large scale, it is difficult to appreciate the differences between groups.

RESPONSE #4: It has been adjusted, accordingly. (Figure 1 to 4)

COMMENT #5: The Authors should include the meaning of the letters a, b, c, and d.

RESPONSE #5: Apologies for the confusion, but we have described the statistical significance of each letter in the figure legend. (i.e., Significant difference exists between groups having different alphabets in superscript (p < 0.05))

COMMENT #6: The Authors should clarify how the samples for the myofibrillar protein synthesis rate were taken, including a description of the method of anesthesia, probe sampling, and care of the animal after the procedure. In case that the samples were taken after had sacrificed the mice, then the Authors should change the word "over' to after. The meaning of over also induces the idea of continuing sampling during the period of 4 weeks.

RESPONSE #6: As suggested, it has been added (Line 145-157). The myofibrillar protein synthesis rate is a 4-week cumulative synthesis rate during the administration of 2H2O over 4 weeks. Thus, “over” is correct.

COMMENT #7: Figure 2: To be able to have a better comparison to other publications and thus further understand the meaning of the parameter integrated protein synthesis, the Authors should include the total muscle weight of all three muscles.

RESPONSE #7: It has been addressed (figure 1).

COMMENT #8: Figure 3: The Authors should run and show the results of an outlier test on the PCR data to check whether these points beyond the distribution of the data can be considered in the analysis.

RESPONSE #8: Regarding outlier, we performed stem-and-leaf plots and boxplots to identify if there were outliers in the PCR data. We found out there are two outliers in the myogenin (one in ACV+EAA and another in ACV+RT+EAA); one outlier in the MuRF1 (ACV+EAA); two outliers in the LC3 (one in ACV+EAA and another in ACV+RT+EAA). Then, we conducted and compared one-way ANOVA on the data with and without outliers to determine whether the outliers have an impact on our results. However, the outlier did not impact on our results, so we remained the outlier in the data. As the data was not normally distributed in the Atrogin-1, we performed Kruskal Wallis test. Both the results that we ran ANOVA and Kruskal Wallis test showed that the mRNA expression of Atrogin-1 was significantly decreased in the ACV+RT, ACV+EAA, and ACV+EAA+RT compared to CON, indicating that outlier in the Atrogin-1 did not impact on our results.

COMMENT #9: Figure 3: The authors should clarify to whom the data was normalized. Moreover, the Authors should show more representative immunoblotting due to the discrepancy between the summarized data and the representative immunoblotting. For instance, it is hard to see that in controls mTORC1 and p-mTOR have the same intensity.

RESPONSE #9: It has been added justified (Figure 3, Line 281, 282).

COMMENT #10: Figure 4: The Authors should clarify what kind of protein is COX IV, why it was selected to analyze mitochondrial protein content, and what was the housekeeping protein used for normalization.

RESPONSE #10: COX IV (cytochrome c oxidase) is the last enzyme of the mitochondrial respiratory chain and it is commonly used as a surrogate marker for mitochondrial content. Thus, we also used COX IV as a marker for mitochondrial content. The housekeeping protein used for normalization has been added Figure 4 (Line 303-304).

COMMENT #11: Figure 4: The Authors should provide information regarding the muscle fiber type before and right after the treatments and muscle fiber diameter. These parameters are classical and relevant to understand the changes induced by the treatments applied here.

RESPONSE #11: We completely understand your point, but unfortunately, further analysis seems to be difficult to perform as there is no remaining muscle tissue.

COMMENT #12: Discussion: The Authors should discuss two papers related to the current manuscript: Effect of mechanistic/mammalian target of rapamycin complex 1 on mitochondrial dynamics during skeletal muscle hypertrophy. Uemichi K, Shirai T, Hanakita H, Takemasa T. Physiol Rep. 2021 Mar;9(5):e14789. doi: 10.14814/phy2.14789. Pretreatment with a soluble activin type IIB receptor/Fc fusion protein improves hypoxia-induced muscle dysfunction. Pistilli EE, Bogdanovich S, Mosqueira M, Lachey J, Seehra J, Khurana TS. Am J Physiol Regul Integr Comp Physiol. 2010 Jan;298(1):R96-R103. doi: 10.1152/ajpregu.00138.2009.

RESPONSE #12: We are grateful that the previous studies have enriched our discussion. We have added it into the discussion based on the study you first suggested that the improvement of muscle quality by functional overload may be at least in part due to changes of mitochondrial fusion and fission to a greater extent than an increase in muscle protein synthesis (Line 388-390). Pistilli et al. in the second study you suggested demonstrated that sActRIIB attenuated the reduction of muscle mass and strength in hypoxia conditions. However, in depth, the magnitude of the reduction in muscle mass and strength in the sActRIIB groups compared to the control appeared to be larger in second week than first week of the hypoxia, which is not comparable with the results in the current study. Therefore, we did not incorporate the paper into the discussion.

COMMENT #13: Line 356: The Authors should explain better the meaning of correlating the 1RM/muscle mass with MuSK expression to give the concept of muscle quality. It is hard to grasp the meaning of this data when the percentage of 1RM/ muscle mass is on the scale of thousands of percent or 1RM/ body mass over hundreds of percent. The physiological question behind this is how the muscle quality (and which quality the Authors want to emphasize with these parameters) is measured with these correlations. Moreover, the Authors should also provide the slope of the correlation to understand the gain of the muscle quality due to the treatment.

RESPONSE #13: We have modified the figure 4 and the discussion. (Figure 4 D, E, Line 93-95, 385-386).

COMMENT #14: Line 361: As the Authors correctly pointed out one of the major limitations of the study, where the muscles were harvested 4 and 11 days after the RT and ACV treatments, respectively. As the Authors also correctly discussed, this large time gap between treatment and tissue harvesting substantially affected the data and thus the interpretation of the results. Therefore, it is necessary then to provide data just after the last treatment to understand the effect of the treatments and comprehend better the physiological effect of such treatment.

RESPONSE #14: As you mentioned, it would strengthen our study if we examine acute response of the treatment. However, it is not possible to conduct additional study regarding acute response of the stimuli as the production of activin receptor provided from our collaborator will take at least several months for the further experiment (if they are willing to do) while the revision process allows us to have only 7 days before re-submission.

COMMENT #15: Another important point discussed by the Authors is the limited number of animals used per study. Whenever possible, the Authors should include another set of animals to see if the Authors can replicate their own data.

RESPONSE #15: Your suggestion was greatly appreciated. Again, in terms of the logistics of time to produce activin receptor and the duration of revision, reconducting the experiment cannot be applicable. And we found similar effects of dietary EAAs on improvement in “muscle quality” in other independent studies.

COMMENT #16: Line 372: the authors should decide whether use branched-chain amino acid or the abbreviation BCAA.

RESPONSE #16: It has been adjusted (Line 405).

Reviewer 3 Report

This is an interesting study with great body of evidence. It is well-written and the figures are well-designed. The following major comments should be addressed by the authors before the paper is suitable for publication.

  1. How did the authors decide how many animals they need for their study? Did they do power calculations?
  2. Please justify the dose of the myostatin inhibitor.
  3. Please justify the duration of the experiment (i.e., 4 weeks).
  4. Why were the mice exercised with and without load? Please explain.
  5. Why did the authors use the specific myostatin inhibitor? Was it based on previous findings?
  6. How is the myostatin inhibition model examined by the authors biologically associated with the molecular mechanisms of sarcopenia, which is mentioned in the conclusion?

Author Response

We appreciate your time and effort in providing these valuable comments.

REVIEWER #3:

This is an interesting study with great body of evidence. It is well-written and the figures are well-designed. The following major comments should be addressed by the authors before the paper is suitable for publication.

COMMENT #1: How did the authors decide how many animals they need for their study? Did they do power calculations?

RESPONSE #1: It has been added (Line 115-118).

COMMENT #2: Please justify the dose of the myostatin inhibitor.

RESPONSE #2: Previously, we examined the difference in the changes of muscle mass by treating 10 or 20 mg/kg of ACV, of which doses are commonly treated in previous ACV studies. Weekly administration of 20 mg/kg in young mice increased muscle mass more efficiently vs. 10 mg/kg. Therefore, based on these results, 20 mg/kg was set as the dose amount for the current study.

COMMENT #3: Please justify the duration of the experiment (i.e., 4 weeks).

RESPONSE #3: Previously, we confirmed that in young mice, ACV treatment increased muscle mass by about 20% over 3 – 4 weeks, but no further increase since then.

COMMENT #4: Why were the mice exercised with and without load? Please explain.

RESPONSE #4: Prior to starting training, the mice underwent to an acclimatization period both to climbing the ladder (exercising without load) and the weight attached to the tail (exercising with load).

COMMENT #5: Why did the authors use the specific myostatin inhibitor? Was it based on previous findings?

RESPONSE #5: sActRIIBfc used in this experiment has already been verified to be effective in increasing muscle mass through previous studies and our pilot study.

COMMENT #6: How is the myostatin inhibition model examined by the authors biologically associated with the molecular mechanisms of sarcopenia, which is mentioned in the conclusion?

RESPONSE #6: While worthwhile, sarcopenia is not our specific focus but muscle wasting in general of the current research. Although not our specific intention, a cross-sectional study of younger, middle-aged and older men suggested that serum myostatin levels increase with advancing age and the myostatin levels are highest in ‘physically frail’ older women, and are inversely associated with skeletal muscle mass (Yarasheski et al., Journal of Nutrition, Health and Aging, 2002). And it was reported in aging mice that inhibition of myostatin suppressed loss of muscle mass (Latres et al., Skeletal muscle, DOI: 10.1186/s13395-015-0060-8).

Round 2

Reviewer 2 Report

The Authors of the manuscript entitled “Myostatin inhibition-induced increase in muscle mass and strength was amplified by resistance exercise training, and dietary essential amino acids improved muscle quality in mice” addressed satisfactory most of all points raised here. However, there are still three major points that the Authors should address before a final decision.

Regarding comment # 5, it is not clear the meaning of the letters above the box plot in each figure. The Authors should explain, for instance, whether the letter “a” is a comparison of control vs. which group. Moreover, it is wrong to add significance in the control group when it is the point of reference to know whether the treatment modifies the analyzed parameters. In other words, the statistical question is always to know whether the treatment modified the analyzed parameters in comparison to the control group. Therefore, it is not clear which comparison was made when the Authors just add an “a”, an “ab”, or a “bc” on top of the data. (comment and answer posted below)

COMMENT #5: The Authors should include the meaning of the letters a, b, c, and d.

RESPONSE #5: Apologies for the confusion, but we have described the statistical significance of each letter in the figure legend. (i.e., Significant difference exists between groups having different alphabets in superscript (p < 0.05))

In comment #13, the Authors state that they address the issue of scale that indicates thousands of percent. This issue is still present in the revised version of the manuscript and it is difficult to understand the physiological meaning of the result. Moreover, it is not answered the question regarding the physiological meaning of muscle quality or Muscle Strength vs MuSK mRNA expression. There are no comments about the physiological relevance of these results in the discussion. The Authors should discuss what is the meaning of such correlations between Hindlimb Muscle Mass to MyoD mRNA expression (Fig. 3) and Muscle Quality and Muscle Strength to MuSK mRNA expression (data from Fig. 4). (comment and answer posted below)

COMMENT #13: Line 356: The Authors should explain better the meaning of correlating the 1RM/muscle mass with MuSK expression to give the concept of muscle quality. It is hard to grasp the meaning of this data when the percentage of 1RM/ muscle mass is on the scale of thousands of percent or 1RM/ body mass over hundreds of percent. The physiological question behind this is how the muscle quality (and which quality the Authors want to emphasize with these parameters) is measured with these correlations. Moreover, the Authors should also provide the slope of the correlation to understand the gain of the muscle quality due to the treatment.

RESPONSE #13: We have modified the figure 4 and the discussion. (Figure 4 D, E, Line 93-95, 385-386).

In the answer for comment # 15, the Authors state that there are “other independent studies” showing the similar effect of dietary EEAs, without referencing the statement in the answer nor in the discussion. (comment and answer posted below)

COMMENT #15: Another important point discussed by the Authors is the limited number of animals used per study. Whenever possible, the Authors should include another set of animals to see if the Authors can replicate their own data.

RESPONSE #15: Your suggestion was greatly appreciated. Again, in terms of the logistics of time to produce activin receptor and the duration of revision, reconducting the experiment cannot be applicable. And we found similar effects of dietary EAAs on improvement in “muscle quality” in other independent studies.

Author Response

RESPONSES TO REVEIWER COMMETNS

We appreciate your time and effort in providing these valuable comments.

REVIEWER #2: The Authors of the manuscript entitled “Myostatin inhibition-induced increase in muscle mass and strength was amplified by resistance exercise training, and dietary essential amino acids improved muscle quality in mice” addressed satisfactory most of all points raised here. However, there are still three major points that the Authors should address before a final decision.

REGARDING COMMENT #5, it is not clear the meaning of the letters above the box plot in each figure. The Authors should explain, for instance, whether the letter “a” is a comparison of control vs. which group. Moreover, it is wrong to add significance in the control group when it is the point of reference to know whether the treatment modifies the analyzed parameters. In other words, the statistical question is always to know whether the treatment modified the analyzed parameters in comparison to the control group. Therefore, it is not clear which comparison was made when the Authors just add an “a”, an “ab”, or a “bc” on top of the data. (comment and answer posted below)

RESPONSE #5: In this paper, it is considered that the concise way to express statistical significances between groups was via alphabets, and it is often used in the previous papers as following references. In addition, legends explaining significance have been modified (i.e., Groups not sharing the same letter are significantly different. (p < 0.05)).

Reference

  1. Phillips et al., Mixed muscle protein synthesis and breakdown after resistance exercise in humans. Am. J. Physiol. 1997, org/10.1152/ajpendo.1997.273.1.E99.
  2. Yang et al., Resistance exercise enhances myofibrillar protein synthesis with graded intakes of whey protein in older men. Br J Nutr. 2012, doi.org/10.1017/S0007114511007422.
  3. Park et al., Metabolic evaluation of the dietary guidelines’ ounce equivalents of protein food sources in young adults: a randomized controlled trial. J Nutr. 2021, org/10.1093/jn/nxaa401.

IN COMMENT #13, the Authors state that they address the issue of scale that indicates thousands of percent. This issue is still present in the revised version of the manuscript and it is difficult to understand the physiological meaning of the result. Moreover, it is not answered the question regarding the physiological meaning of muscle quality or Muscle Strength vs MuSK mRNA expression. There are no comments about the physiological relevance of these results in the discussion. The Authors should discuss what is the meaning of such correlations between Hindlimb Muscle Mass to MyoD mRNA expression (Fig. 3) and Muscle Quality and Muscle Strength to MuSK mRNA expression (data from Fig. 4). (comment and answer posted below)

RESPONSE #13: As the muscle weight at the whole-body level was not available in the current paper, the maximal carrying capacity (g) was normalized to hindlimb muscle weight (mg). For that reason, the scale was expressed in thousands of percent (e.g., 80g of maximal carrying capacity / 0.8 g of hindlimb muscle mass = 10,000%) (Fig.4 D). Likewise, muscle strength is expressed in hundreds of percent (e.g., 80g of maximal carrying capacity / 24 g of body weight = 333%) (Fig.4 E). The physiological meaning of correlations between hindlimb muscle mass and MyoD mRNA expression (Line 358, 360-361) and between the muscle quality and muscle strength to MuSK mRNA expression (Line 386-387, 389-391) has been modified.

IN THE ANSWER FOR COMMENT #15, the Authors state that there are “other independent studies” showing the similar effect of dietary EEAs, without referencing the statement in the answer nor in the discussion. (comment and answer posted below)

COMMENT #15: Another important point discussed by the Authors is the limited number of animals used per study. Whenever possible, the Authors should include another set of animals to see if the Authors can replicate their own data.

RESPONSE #15: Further experiments are not feasible mainly due to time (> 3 months) required for generating new batch of the myostatin inhibitor. And we think that the EAA effect is not by chance as we observed the similar effect in another independent study for other purposes, whose data will be submitted later for publication.